# Fabrication of Li-Doped NiO Thin Films by Ultrasonic Spray Pyrolysis and Its Application in Light-Emitting Diodes

**DOI:** 10.3390/nano13010197

**Published:** 2023-01-01

**Authors:** Víctor Hugo López-Lugo, Manuel García-Hipólito, Arturo Rodríguez-Gómez, Juan Carlos Alonso-Huitrón

**Affiliations:** 1Instituto de Investigaciones en Materiales, Universidad Nacional Autónoma de México, Ciudad Universitaria, A.P. 70-360, Coyoacán, Mexico City 04510, Mexico; 2Instituto de Física, Universidad Nacional Autónoma de México, Ciudad Universitaria, A.P. 20-364, Coyoacán, Mexico City 04510, Mexico

**Keywords:** nickel oxide, Li-doped, thin films, ultrasonic spray pyrolysis, heterojunction LED, electroluminescence

## Abstract

The fabrication of NiO films by different routes is important to extend and improve their applications as hole-transporting layers in organic and inorganic optoelectronic devices. Here, an automated ultrasonic pyrolysis spray method was used to fabricate NiO and Li-doped NiO thin films using nickel acetylacetonate and lithium acetate dihydrate as metal precursor and dimethylformamide as solvent. The effect of the amount of lithium in the precursor solution on the structural, morphological, optical, and electrical properties were studied. XRD results reveal that all the samples are polycrystalline with cubic structure and crystallite sizes in the range of 21 to 25 nm, without any clear trend with the Li doping level. AFM analysis shows that the crystallites form round-shaped aggregates and all the films have low roughness. The optical transmittance of the films reaches values of 60% to 77% with tendency upward as Li content is increased. The electrical study shows that the films are p-type, with the carrier concentration, resistivity, and carrier mobility depending on the lithium doping. NiO:Li (10%) films were successfully incorporated into inorganic light emitting diodes together with Mn-doped ZnS and ZnO:Al films, all deposited on ITO by the same ultrasonic spray pyrolysis technique.

## 1. Introduction

Non-stoichiometric nickel oxide (NiO_x_) is a wide band-gap p-type semiconductor with electrochromic, electrochemical, electrical, and optical properties adequate for a wide range of applications. Pure and doped NiO thin films can be used for the fabrication of H_2_ and NO_2_ gas sensors [1,2,3] and electrochromic devices [4,5]. NiO_x_ thin films have also been successfully applied as hole transporting layers (HTLs) in organic and perovskite solar cells [6,7,8,9,10], and in inorganic and perovskite light-emitting diodes (LEDs) [11,12,13,14,15,16,17,18]. Various methods can be used to prepare NiO thin films, such as RF and DC magnetron sputtering [4,19,20], electron-beam evaporation [21], pulsed laser deposition [22,23], atomic layer deposition [24], metal-organic chemical vapor deposition [25], chemical bath deposition [26], spin coating [27], sol-gel process [28], SILAR [29], and Spray Pyrolysis (SP) [30,31]. Among all these methods, the SP technique has the advantage of being easy and economical because it does not need high vacuum equipment and offers the possibility for large area deposition. Other advantages of SP are the diversity of its solution precursor and the higher stability of the obtained films as compared to films deposited in vacuum [32]. For the applications as HTLs the NiOx films have been generally deposited on indium tin oxide (ITO) coated glass substrates, mostly by spin coating methods, using different nickel precursors solutions and posterior annealing processes at different temperatures and ambient conditions [12,13,14,15,18]. Recently, densely packed NiO films were prepared on pre-cleaned ITO glass substrates by using magnetron sputtering and a high-purity NiO (99.999%) ceramic target for the fabrication of high-efficiency deep-blue light-emitting diodes [16]. Since the resistivity of NiO can be decreased by doping with monovalent atoms like Li, Li-doped NiO thin films have also been deposited by techniques such as laser deposition, spin coating, and SP [33,34,35,36,37,38,39,40]. In the case of NiO:Li films deposited by SP, different precursor solutions have been used, and the transmittance and electrical properties of the films have been investigated as a function of the main deposition conditions, such as, concentration of Ni and Li precursors in the precursor solution and substrate temperature. For NiO:Li films deposited by pneumatic SP from nickel nitrate hexahydrate (Ni(NO_3_)_2_·6H_2_O) and lithium chloride (LiCl) dissolved in distilled water [37], the highest grain size and hole concentration (7.11 × 10^15^ cm^−3^) is obtained for films deposited with the highest concentration of the nickel precursor (0.2 M) in solution., but for this concentration the transmittance is reduced to values lower than 60% for wavelengths lower than 600 nm, and the incorporation the Ni_2_O_3_ phase in the films is observed in the diffraction patterns. The concentration of holes also increases up to 1.14 × 10^18^ cm^−3^ as the LiCl precursor concentration in solution increases up to 30%, but for higher concentrations the hole concentration and transmittance of the films is drastically decreased, and NiCl_2_ phases in the films are observed. A similar trend is found for NiO:Li films deposited by a modified SP method from nickel and lithium nitrate dissolved in deionized water and isopropyl alcohol and annealed at 600 °C for densification and crystallization [38]. In this case, the highest hole concentration (3 × 10^18^ cm^−3^) is obtained for films deposited with the highest concentration of the Li precursor (10%) in solution, but for this concentration the transmittance is reduced to values lower than 60% in all the visible range. A close work found that the hole concentration of NiO:Li films prepared with the same modified SP method and nitrate precursors, but with 8% of the Li precursor in solution [39], increases from 2 × 10 ^17^ cm^−3^ to 1 × 10^18^ cm^−3^ after they are annealed from 400 to 600 °C, respectively, in air for three hours, and the transmittance increase to values higher than 80% for wavelengths higher than 500 nm. However, the roughness of these films is high (17 nm). NiO:Li films with very low roughness (0.33–0.5 nm) and with transmittance around 80% have also been deposited by SP using nickel and lithium chloride dissolved in redistilled water, but in this case the electrical properties were not reported [40]. It is worth mentioning that only in one of the previous works the spray deposited NiO:Li films were applied to the fabrication of a heterojunction diode, but any of these films has been applied as a hole transporting layer in an electroluminescent device. 

In this work, p-type Li-doped NiO thin films have been prepared by ultrasonic spray pyrolysis (USP), using Nickel (II) acetylacetonate and lithium acetate dihydrate dissolved in N, N. Dimethylformamide (DMF). The choice of these precursors and solvent was based on previous reports showing that dense thin films, with excellent quality of metal oxides, such as aluminum oxide, yttrium oxide and zirconium oxide, can be deposited by SP using metal acetylacetonates and DMF as the only solvent [41,42]. DMF is considered as a universal solvent and has the advantage of having high dielectric constant and high boiling point. On the other hand, the USP method has the advantage over the pneumatic SP method of having a narrower distribution of drops with smaller size which renders in a better control of the morphology and quality of the films [32,41]. Based on these advantages we obtained NiO:Li films with low roughness, good transparency and electrical properties, which were successfully applied for the fabrication of inorganic light emitting diodes, where all the other films of this device were also deposited by the automated USP method. 

## 2. Materials and Methods

The NiO and NiO:Li thin films were deposited by an automated USP technique at atmospheric pressure, using a low-cost homemade spray system with an Mxmoonant ultrasonic mist maker of 1.7 MHz vibration frequency. The glass bell aerosol nozzle assembly was mounted on an adapted Arduino CNC router to precisely control (with >0.1 mm accuracy) the nozzle’s trajectory and speed respect to the substrate, and the number of deposition cycles. With this system and under the deposition conditions used in this work we obtain uniform thin films deposited over substrates with 1 inch × 1 inch area. The starting solution for the deposition of the NiO thin films consisted of 0.2 M of Nickel (II) acetylacetonate (95%, Sigma-Aldrich—Merck Group, St. Louis Missouri, USA) dissolved in N, N. Dimethylformamide (99.9%, Sigma-Aldrich—Merck Group, St. Louis Missouri, USA). For the impurification with Li, lithium acetate dihydrate (98%, Sigma-Aldrich—Merck Group, St. Louis Missouri, USA) was added to the starting solution in different molar percentages (5, 10, 15 and 20%). The films were deposited at atmospheric pressure on glass substrates heated on a tin bath at a temperature of 450°C. We used high purity dry nitrogen as carrier and director gas at fixed flow rates of 6.2 l/min and 0.65 l/min, respectively. For the deposition of each film, we used the same volume (22 mL) of the starting solution. All the films were deposited in a time of 25 min. For the characterization, most of the films were deposited on 2.5 × 2.5 cm^2^ corning glass slices, which were cleaned before film deposition, using an ultrasonic bath, first with trichloroethylene, then with acetone and finally with methanol. For the fabrication of the heterostructure light emitting diodes, ITO coated (135 nm, 10–15 Ohm/sq) glass substrates (Biotain Hong Kong Co., Limited, Xiamen, Fujian, China) were used. For the deposition of the NiO:Li (10%) films on ITO substrates, the conditions were the same as for the case of the films deposited on glass. The structural analysis of the synthesized NiO:Li (x %) thin films deposited on glass and ITO was done using a Bragg–Brentano Rigaku ULTIMA IV diffractometer (Rigaku Corporation, Austin, TX, USA) with an X-ray source of Cu Kα line (0.15406 nm), at a grazing beam configuration (incidence angle of 1°). The surface morphology of the films was investigated by using atomic force microscopy (AFM) for this purpose JSPM-4210 scanning probe microscope and scanning electron microscopy (SEM), JEOL JSPM-4210 scanning probe microscope were used. The optical transmittance of the films was measured, in the range from 190 to 1100 nm, using a double beam PerkinElmer 35 UV-vis spectrophotometer (Waltham, Massachusetts, USA). The refractive index of some films deposited on silicon was measured using a Gaertner L117 ellipsometer equipped with a He-Ne laser (λ = 632.8 nm) at an incidence angle of 70°. 

## 3. Results and Discussion

### 3.1. Structure and Morphology

Figure 1 shows the XRD patterns for a series of films deposited at different concentrations of Li (0–20%). According to the card number 01-080-5508 from pdf NiO (2022) data base the deposited films are polycrystalline and present a cubic structure. All of them show two mains diffraction peaks indicating a preferential growth in the planes: (2 0 0) and (2 2 0). Other planes such as: (1 1 1) and (3 1 1) are also present, but their intensities are very small. As observed in Figure 1 no peaks of any other phase of NiO or impurity peaks are observed, which indicates the high purity of the NiO obtained. Besides, the peaks of diffraction are sharp, narrow, and symmetrical with a low and stable baseline, suggesting that the samples are well crystallized.

Since the films present a cubic structure, the lattice parameter (a) can be obtained from the equation [43]:(1)$$a=d_{hkl}\sqrt{h^{2}+k^{2}+l^{2}}$$
where a is the lattice parameter, $d_{hkl}$ is the inter-plane spacing, and h, k, l are the Miller indices of the planes. The inter-plane spacing can be calculated using the Bragg formula [43]:(2)$$d=\frac{\lambda }{2\sin\theta }$$
where λ is the X-ray wavelength (0.1540 nm) and θ is the diffraction angle. The crystallite size D was calculated using the Sherrer’s formula [40,43,44]:(3)$$D=\frac{k\lambda }{\beta \cos\theta \;}$$
where k is the correction factor equal to 0.9, β is the full width at half maximum (FWHM) and θ is the diffraction angle.

The position of the peaks associated with the (2 0 0) plane, the inter-plane spacing, the lattice parameter and the grain size for every sample are shown in Table 1. 

As can be seen in this table, the lattice constant values of the nickel oxide films deposited with different lithium concentrations are very close to that reported with standard value: 0.417–0.418 nm [35,36,43]. The calculated crystallite sizes were in the range from 21.7 nm to 25.2, without any clear trend in size as the concentration of the dopant increases. This indicates that Li doping does not have any significant influence on the crystallite size, which is an expected result if the Li ions effectively substitute the Ni ions, since the effective ionic radius of Li^+^ (0.76 Å) is close to the effective ionic radius of Ni^2+^ (0.69 Å) [45]. This is in contrast to what has been published in other works such as [36,46] where crystalline size of the nickel oxides decreases as the lithium concentration increases. The actual Li concentration in the films was not measured. However the fact that no any traces of Li_2_O in the XRD patterns, and the close structural parameters of NiO:Li films, as compared with the NiO film, indicates that the incorporation of Li is at the doping level and well below the solubility limit [47]. This is consistent with some previous specific works on spray pyrolysis deposition which show that Li incorporated in NiO films is lower (~5%) than that provided in the start solution (20%) [48].

The surface morphology of undoped and lithium doped films were studied by AFM. To evaluate the surface roughness, an area of a 1 μm × 1 μm has been scanned in tapping mode. The bi-dimensional AFM images are shown in Figure 2. All the samples have a uniform distribution and dense round-shaped grains whose average size varies from ~ 150 nm to ~ 90 nm as the Li concentration is increased. The larger sizes of the grains compared with the crystallites size calculated and shown in Table 2, indicates that during the growth of the films the small crystallites aggregate to form the larger polycrystalline grains. The RMS roughness of the thin films increases from 5.47 nm for the NiO film without Li up to 6.40 nm for the NiO:Li film deposited with a 20% of the lithium precursor in the solution. 

Figure 3a shows typical SEM image of the NiO: Li (10%) film deposited on glass. The film exhibits agglomerates of round and cubic shaped and well packed agglomerates, with average size 100 nm, which is consistent with that observed in the AFM images of the other samples. The regular and highly orientated structure agrees with the preferential orientation shown in the XRD diffractogram for this film. Figure 3b shows a cross section for the same film, it exhibits excellent adhesion, as well as a flat and compact surface with a thickness of 450 nm.

### 3.2. Optical and Electrical Properties

Figure 4 shows the optical transmittance of the NiO films deposited on glass substrates with different Li concentrations. The transmittance spectra were measured with the light beam incident by the side of the film and at the center of the samples. The transmittance of the glass substrate on which the films were deposited, it is also shown in Figure 4. The average transmittance was calculated in the range from 400 to 800 nm. As can be seen in Figure 4 and Table 2, the transmittance of the films gradually increases from 60% to about 80% in the visible region as lithium is increased. It is also observed that as the dopant increases, the number of interference fringes decrease due to a decrease in the thickness of the film. 

The accurate calculation of the thickness and optical constant of a thin layer deposited on a transparent substrate, from the experimental transmittance spectrum is a very challenging problem. Some of the most accurate methods for the determination of the thickness and optical parameters of a thin film have been developed for amorphous silicon films, and these are based on the rigorous expression for the optical transmission of the system of a thin absorbing film on a thick finite transparent substrate [49,50]. In our case, the thickness and optical constants of the NiO:Li films were calculated using a similar rigorous method recently applied for transparent conducting coatings of ZnO:Al deposited on glass substrates by ultrasonic spray pyrolysis [51]. The model is based on the Drude-Lorentz model and Sellmeier expressions for the real and imaginary parts of the dielectric function and considers the Urbach- tail absorption edge at the low wavelength region, the contribution of free carrier concentration to the weak absorption in the visible and near-infrared ranges, and the effect of scattering of light originated by the surface roughness of the films. Under this model the theoretical expression for the transmittance is given by the Equation (12) in reference [51], which is: (4)$$T_{Model}=\left[\frac{T_{3}}{1-R_{2}^{’}R_{3}^{’}}\right]\left(\frac{T_{1rs}T_{2rs}e^{-\alpha d}}{1-2R_{1rs}^{1/2}R_{2rs}^{1/2}cos\Phi +R_{1rs}R_{2rs}e^{-2\alpha d}}\right)$$
where d is the thickness of the film, α is the absorption coefficient, and the subindex in the transmittance and reflectance coefficients refer to the respective interfaces (1: air-film, 2: film-substrate, 3: substrate-air). The term with the factor, cosΦ, where Φ= $\frac{4\pi }{\lambda }n\left(\lambda \right)d$, and n(λ) the real part of the refractive index of the film, gives rise to the maxima and minima of interference. The dependence of the refractive index with wavelength is obtained from a given model, through the well know interrelationships between the respective real and imaginary parts of the complex index of refraction and the real and imaginary parts of the complex dielectric function [50]. The details of the fitting process between the theoretical transmittance and the experimental transmittance spectra are out of the scope of this work and will be published elsewhere. Here we only present in Figure 5 and Table 2, the results of the best fitting from which the thickness and energy band gap of the films were determined. The value of the mean squared error (MSE) for each fitted spectrum is also shown in Figure 5. It is worth to mention that the minimal MSE was obtained for the NiOLi (10%) film, with a calculated thickness of 194 nm, which is very close to the thickness (197 nm) measured by SEM cross section for the same film deposited onto the ITO coated glass substrate. It is also important to mention that the behavior of the modeled refractive index n(λ) used for the fitting, resulted similar to that reported in reference [23], and the value of n(λ=632 nm)=2.35 is consistent with the average refractive index measured (n=2.3±0.2) for some of the films deposited on silicon substrates. 

Table 2 shows that the band gap increases from 3.3 to 3.7 eV with increasing lithium content in the films. High values of the band gap can be explained by the fact that the films are deposited at high temperatures as well as the increase in the concentration of lithium, two factors that contribute to the improvement in crystallinity and less stoichiometric defects as it has been reported in works as [34].

The electrical resistivity, carrier concentration and mobility of the films were measured at room temperature using Hall measurements by the four-point van der Pauw method, using an Ecopia HMS-3000 system (Ecopia Corp., Anyang City, Gyeonggi-Do, Republic of Korea). The measurements carried out on all the films exhibited a positive Hall coefficient, confirming its p-type conductivity. Table 3 shows the electrical parameters obtained for all the samples, except for the NiO: Li (0%) films which is highly resistive to be measured, since it has been reported that the resistivity of undoped NiO can reach the value of $10^{6}$ Ohm-cm [46], indicating that there is no a significant density of electrically active defects, that is, Ni vacancies (VNi) [35]. As can be observed from the data of Table 3, the carrier concentration increases as the amount of lithium in the precursor solution increases from 5% and 20%. This is due to the increase of the hole density in the nickel oxide lattice as the $Li^{+}$ ions occupy the $Ni^{2+}$ [36]. 

The increase in charge carriers is explained through the mechanism of incorporation of Li into the NiO lattice that consists of two steps: first lithium enters the lattice substitutionally creating oxygen vacancies and then the lattice takes oxygen from the surroundings [36,52]. The carrier mobility also increases Li doping increases, however the maximum hole mobility ($2.9\;\frac{cm^{2}}{V\cdot s}$) is obtained for the NiO:Li(10%) sample, and then the further increase in lithium over the NiO:Li( 10%) sample negatively affects the carrier mobility, and it is decreased. This can be explained by the dispersion effect of the ionized impurities as well as the presence of neutral defects due to the excess of Li incorporated in the NiO lattice, which reduce the mobility of the charge carriers [53].

### 3.3. NiO: Li in a LED Structure

We also have investigated the suitability of our p-type NiO:Li films as a hole transporting layer by fabricating a LED structure. For this purpose, we have chosen NiO: Li (10%) because they presented the highest mobility and therefore better hole carrier injection properties. The LED fabricated was a heterojunction ITO/NiO:Li (10%)/ZnS: Mn^2+^ (10%)/ZnO: Al (3%), all deposited by the automated USP technique on an ITO-coated glass substrate as bottom conducting cathode. As an alternative to the LEDs that have been manufactured from (GaAs) it is proposed to incorporate more accessible materials in terms of costs and abundance on the planet. In this case NiO:Li (10%) was used as an electron blocking and hole transport interlayer with the aim of avoiding non-radiative centers by controlling the region of the recombination luminescence area. It has been reported in other works that it has significantly improved the efficiency in the emission of LEDs, hence the interest in the use of this thin film to incorporate it in this research [12,18]. The ZnO with wurtzite structure is one of the promising materials with wide band gap (3.37 eV) which has been widely considered as a high potential material for use in optoelectronic devices [54]. Specifically, light emitting diodes and laser diodes based on ZnO material have been explored for their usability. For our LED structure, an n-type ZnO:Al thin film was deposited at low temperatures ( 360 °C) using the USP technique and the same precursor solution and conditions reported in a previous work [44]. This layer was used as electron injection layer due to their low electrical resistivity $\left(∼10^{-3}\;Ohm-cm\right)$, high concentration of charge carriers ($∼10^{20}\;cm^{-3})$ and high optical transmittance. One of the most studied materials in terms of emission properties and the possibility of including them in light-emitting diodes and other electroluminescent devices, has been manganese-doped zinc sulfide (wurtzite structure). Zinc sulfide has been suitable as a semiconductor to be used as a host matrix for a wide variety of rare earth and transition metal luminescent centers this is explained by the fact that the crystalline matrix has a wide forbidden gap of 3.77 eV [55]. In this work, the ZnS:Mn thin film for the LED structure was also deposited in the USP system under deposition conditions reported in a previous work [56]. The thickness of this film was 450 nm, as it was measured using a Dektak IIA profilometer. As it is known these thin film present a yellow-orange emission peak when Mn ^+ 2^ is incorporated substitutionally into the lattice ZnS, and whose emission is caused by the transition of levels ^4^$T_{1}$(^4^G)→ ^6^$A_{1}$(^6^S), corresponding to a forbidden transition so that there is an emission more intense than that of centrosymmetric systems [57]. The schematic diagram of the fabricated light-emitting diode based on the ITO/NiO:Li/ ZnS:Mn^2 +^ / ZnO:Al heterojunction is shown in Figure 6. The active area of our DC driven devices was determined by the radius (1 mm) of the circular Al electrodes. The aluminum (purity of 99.999%) electrodes with a thickness of 100 nm were thermally evaporated (5–6 × 10^−6^ Torrs) using a shadow mask.

Figure 7a depicts the SEM image for the film NiO: Li (10%) deposited on ITO substrates, in this case, it can be observed grains less rounded with some edges and larger than those observed in Figure 3a, for the same film deposited on glass. Figure 7b shows a cross section SEM micrograph of the same NiO: Li (10%) film. As can be seen the film appears well adhered to the ITO film, in addition the film shows a uniform thickness of approximately 197 nm. Figure 7c shows the diffractogram of the film deposited on the substrate covered with ITO, in which one main diffraction peak is presented indicating a preferential growth in the plane: (2 0 0 ). Other planes such as: (1 1 1), (2 2 0) and (3 1 1) are also presented but in reduced intensity, the peaks marked with * correspond to those of the ITO. The grain size associated with the plane (2 0 0) was calculated using equation 3 that corresponds to 20.21 nm a value not very far from that obtained for the case of the film deposited on glass. The value of the lattice constant was 0.4165 nm obtained from Equation (1). 

To record the current voltage (I-V) characteristics of the LED, a Keithley 2450 dual channel System SourceMeter Instrument was used. Positive bias was applied to ITO electrode with respect to the top one of Al. The formation of NiO: Li /ZnO:Al p-n junction with ZnS: Mn as phosphorescent interlayer was confirmed by the I-V characteristics. As can be observed in Figure 8a the LED structure demonstrates non-linear rectifying behavior. The threshold voltage of the p-n junction, under which electroluminescence is observed, is about 4.0 V under forward bias. No electroluminescence was observed under reverse bias conditions. As shown in that figure, the fitting equation is a direct proportionality relationship between I and $V^{2}e^{\left(\frac{a}{V}\right)}$ which is governed by the Fowler-Nordheim equation that it has been used in quantum dot based sandwiched structures to explain tunneling current [12,56]. Given the barrier for the electrons between the ZnO:Al conduction band and the ZnS:Mn conduction band, under forward biasing the conduction band of ZnS:Mn is tilted and the model of a tunneling current through a triangular barrier is well applied. Then, there is electron acceleration by the electric field inside the ZnS:Mn and impact excitation of the Mn^2+^ luminescent centers. The forward bias turn-on voltage of the heterostructure ITO/NiO:Li/ZnS:Mn/ZnO:Al is relatively high, which is attributed to the high barrier heights at the interfaces [58] as can be observed in Figure 8b. 

Figure 9 shows the photoluminescent spectrum of the ZnS:Mn film used for the LED fabrication and the electroluminescent spectrum of the LED operated at 9.5 V.

Both, the PL and EL spectra present a peak emission around 582 nm characteristic of radiative transitions between the d levels of the Mn^2+^ ion. The EL spectrum presents a very narrow peak width which indicates that the emission is very localized and that the emission times could be very short, another factor that could affect the width of the EL curve is the heating of the sample, due to the fact that not all the powered energy is directed towards the displacement of the electrons but part of it is converted into phonons in the crystal lattice.

## 4. Conclusions

P-type Li doped NiO thin films were obtained by the ultrasonic spray pyrolysis technique. The use of nickel acetylacetonate and lithium acetate dihydrate as metal precursor and dimethylformamide as solvent, allowed the fabrication of smooth and flat polycrystalline NiO:Li films with good optical transmittance in the visible range and good electrical properties for its application as hole transporting layer in an optoelectronic device. The sample NiO:Li (10%) with the highest hole mobility was used to be incorporated into an LED structure serving as a hole-transporting and electron-blocking layer at the same time. The I vs V curve for the ITO/NiO:Li/ZnS:Mn/ZnO:Al structure was obtained showing a non-linear rectifying behavior. The EL and PL spectra were obtained and compared; in both cases they presented an emission around 585 nm due to the transitions between the d states of the Mn^+2^ ions. It should be noted that most of the elements used in the incorporation of the LED are abundant elements on earth and are non-toxic.

## Figures and Tables

**Figure 1 nanomaterials-13-00197-f001:**
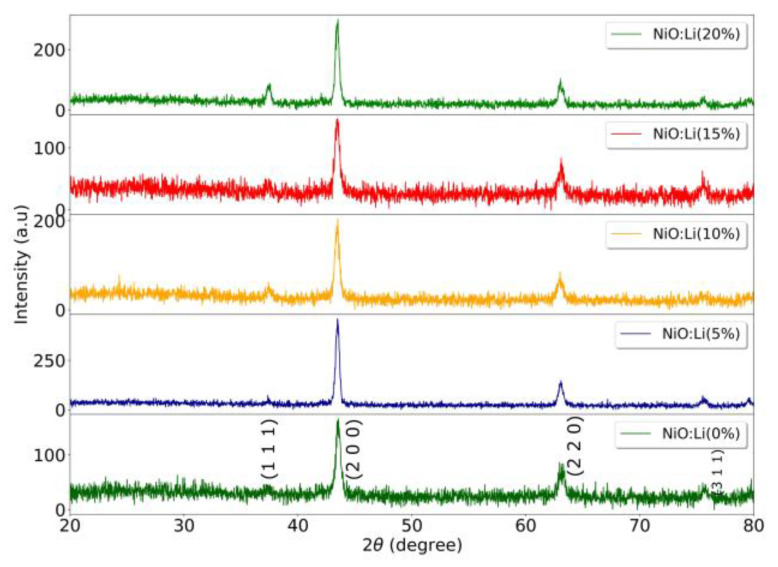
XRD patterns of NiO:Li films deposited with different lithium concentrations in the precursor solution.

**Figure 2 nanomaterials-13-00197-f002:**
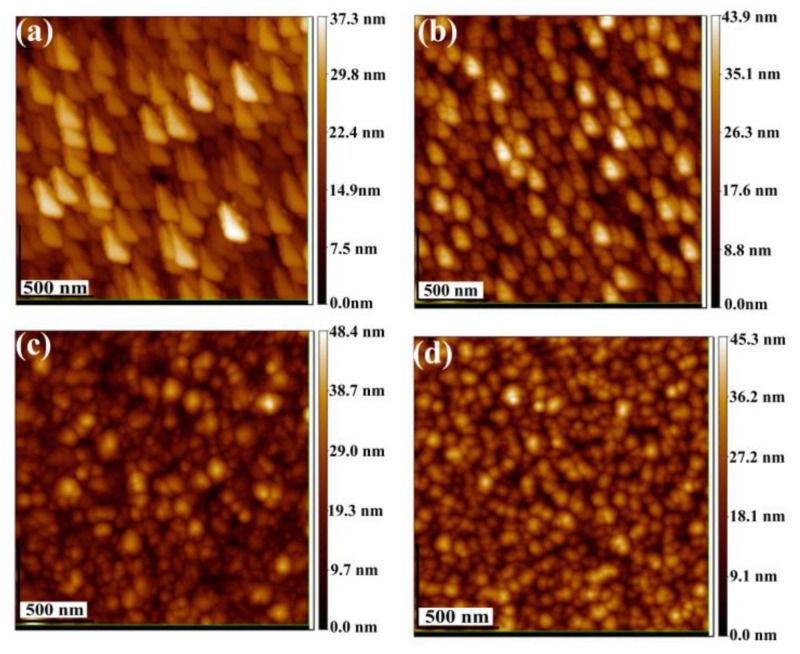
AFM images of undoped and lithium doped NiO thin films. (**a**) NiO: Li (0%); RMS = 5.47 nm, (**b**) NiO: Li (5%); RMS = 5.97 nm, (**c**) NiO: Li (15%); RMS = 6.29 nm and (**d**) NiO: Li (20%); RMS = 6.40 nm.

**Figure 3 nanomaterials-13-00197-f003:**
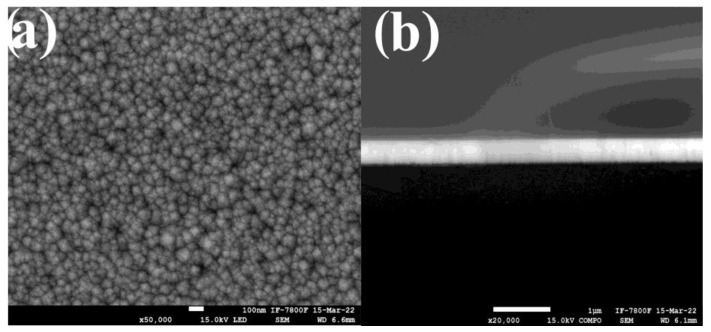
(**a**) SEM micrograph for NiO: Li (10%) deposited on glass. (**b**) SEM Cross section image of the same film.

**Figure 4 nanomaterials-13-00197-f004:**
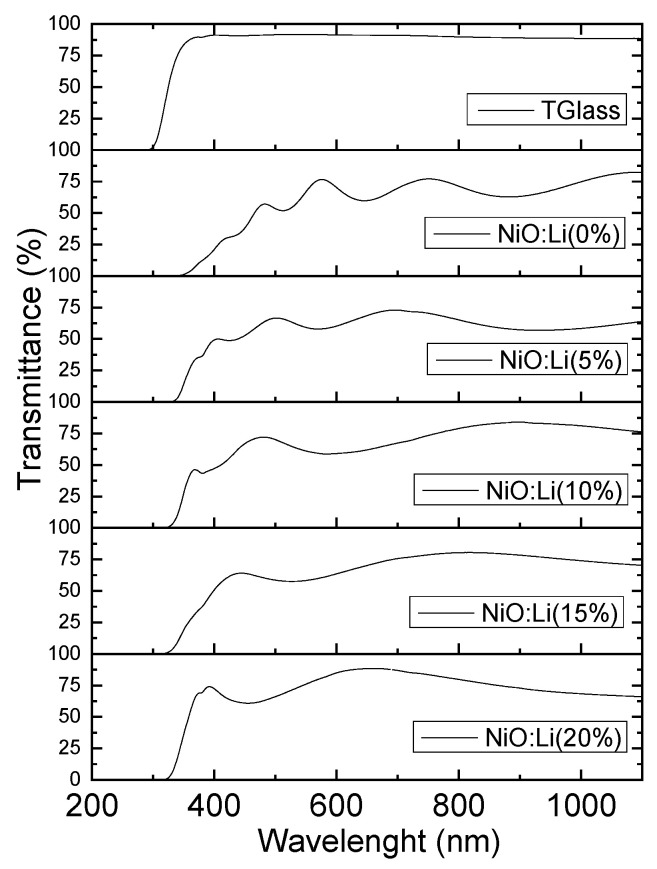
Optical transmittance spectrum for NiO:Li (x%) films deposited at different lithium concentrations. The spectrum at the top is the transmittance of the glass substrate.

**Figure 5 nanomaterials-13-00197-f005:**
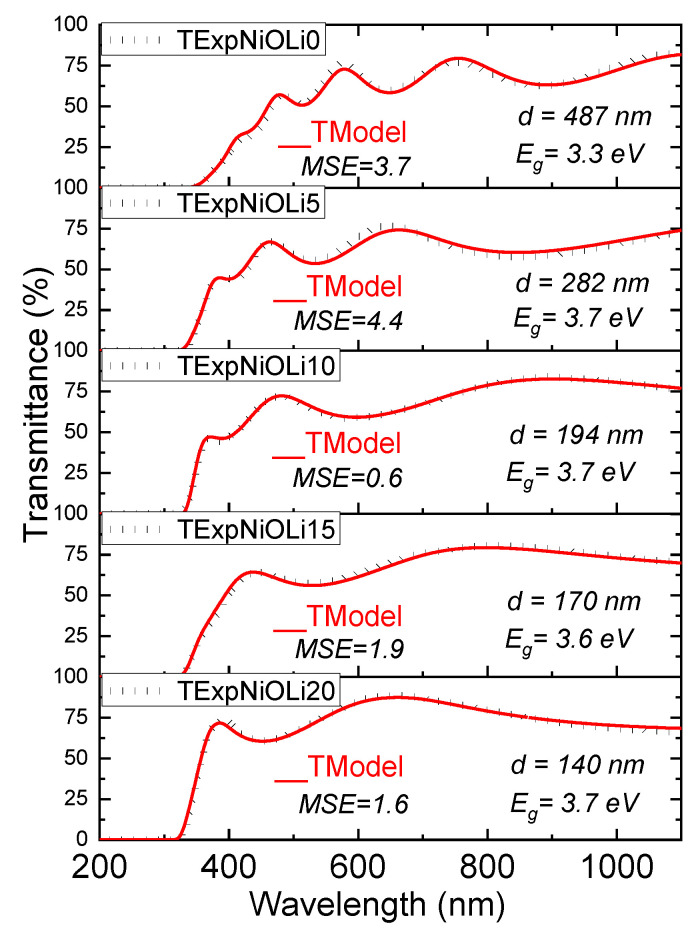
Experimental transmittance spectra and the best modeled transmittance for the NiO films with the different concentrations of Li in the precursor solution.

**Figure 6 nanomaterials-13-00197-f006:**
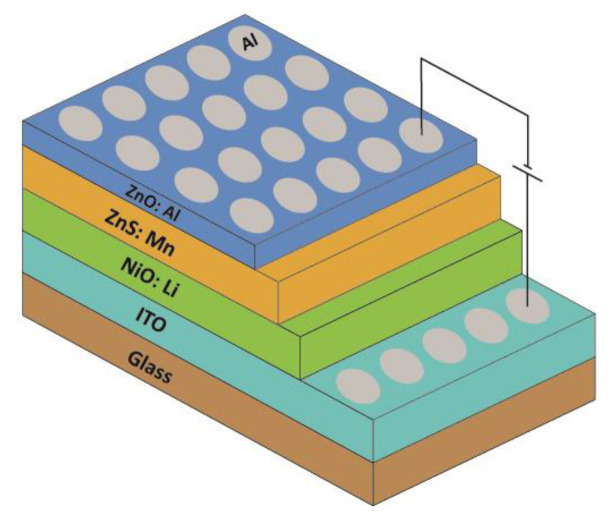
Schematic diagram of fabricated LED.

**Figure 7 nanomaterials-13-00197-f007:**
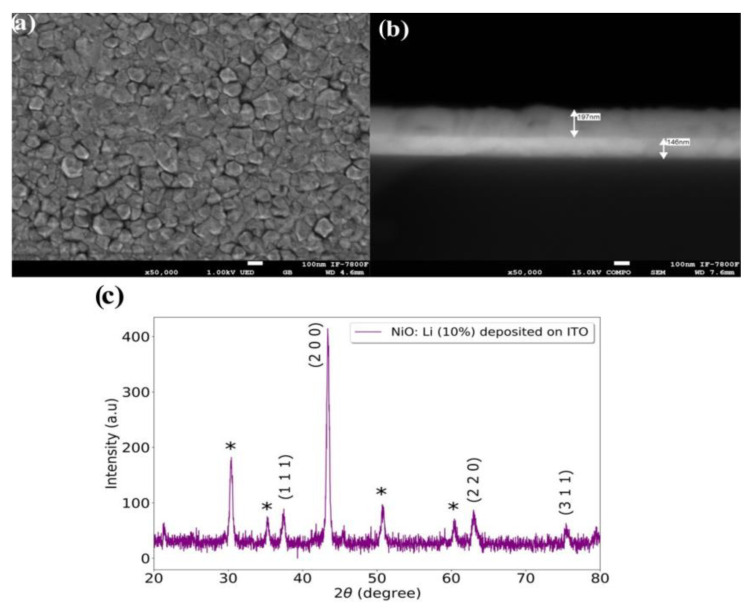
SEM images and XRD pattern of NiO: Li (10%) film deposited on ITO coated glass substrate. (**a**) SEM image. (**b**) Cross section SEM micrograph. (**c**) XRD pattern, it shows a cubic structure and preferential orientation in the plane (2 0 0). The diffraction peaks marked with * correspond to ITO.

**Figure 8 nanomaterials-13-00197-f008:**
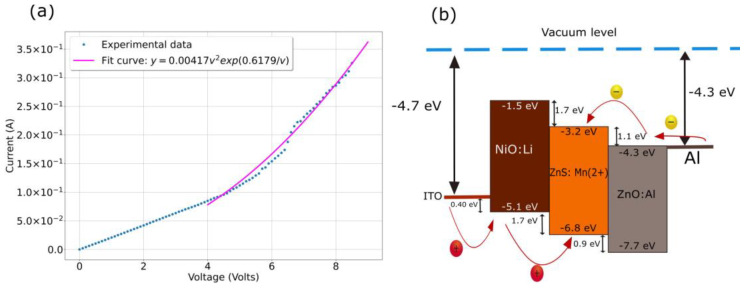
(**a**) I-V curve of the device with NiO:Li intermediate layer. (**b**) Band diagram of ITO/NiO:Li/ZnS:Mn/ZnO:Al sandwiched structure.

**Figure 9 nanomaterials-13-00197-f009:**
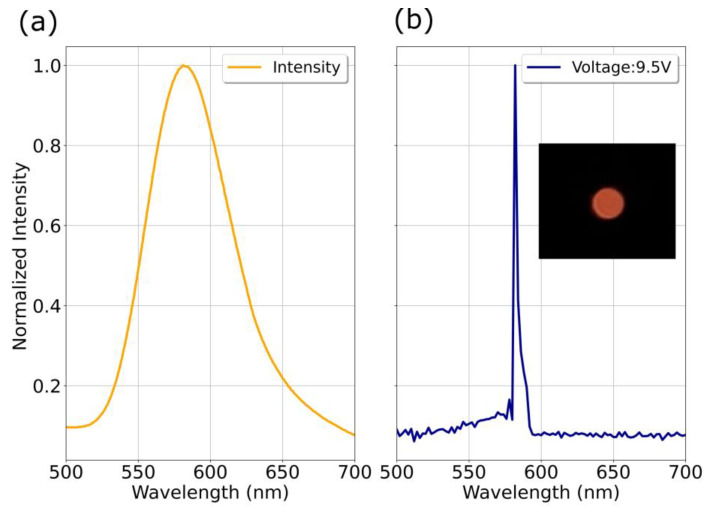
On the left side (**a**), the photoluminescent spectrum is shown whose emission peak corresponds to about 582 nm. On the right side (**b**) the electroluminescent spectrum of the LED device under an operating voltage of 9.5 V is shown. The inset shows a photograph of the light emitted by the device.

**Table 1 nanomaterials-13-00197-t001:** Structural parameters of undoped and lithium doped NiO thin films.

Film	2θ(°)	d Value (Å)	Grain Size (nm)	a (Lattice Constant) (Å)
NiO (20%)	43.446	2.081	24.3	4.162
NiO (15%)	43.450	2.081	21.7	4.162
NiO (10%)	43.442	2.081	22.6	4.163
NiO (5%)	43.475	2.080	25.2	4.160
NiO (0%)	43.540	2.077	22.1	4.1538

**Table 2 nanomaterials-13-00197-t002:** Average transmittance in the visible region, and thickness and band gap energy of the films determined from the transmittance fitting model.

Film	Average Transmittance (%)	Thickness (nm)	Eg(eV)
NiO:Li (20%)	77.6	140	3.7
NiO:Li (15%)	67.1	170	3.6
NiO:Li (10%)	65.4	194	3.7
NiO:Li (5%)	63.8	282	3.7
NiO:Li (0%)	60.9	487	3.3

Film

**Table 3 nanomaterials-13-00197-t003:** Electrical properties of NiO:Li films correspond to different concentrations of Li in the precursor solution.

Film	*ρ* (*Resistivity*) [*Ohm—cm*]	$$n_{p}\left[cm^{-3}\right]$$	$$\mu \left[\frac{cm^{2}}{V\cdot s}\right]$$
NiO: Li (0%)	---------------	-------------------	--------------------
NiO: Li (5%)	2.5 × 10^3^	3.3 × 10^14^	1.6
NiO: Li (10%)	4.4 × 10^2^	4.9 × 10^15^	2.9
NiO: Li (15%)	2.8 × 10^2^	2.1 × 10^18^	1.1
NiO: Li (20%)	2.3 × 10^2^	1.4 × 10^17^	0.2

## Data Availability

Not applicable.

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
