# Peer review of "Fabrication of Li-Doped NiO Thin Films by Ultrasonic Spray Pyrolysis and Its Application in Light-Emitting Diodes"

_nanomaterials, 2023, doi:10.3390/nano13010197_

Round 1
Reviewer 1 Report
In the submitted manuscript the Authors discuss on the fabrication and physical properties of NiOx films made through ultrasonic spray pyrolysis technique. A series of characterizations like morphology, transport measurements and photonic absorption are conducted to assess the use of NiOx as hole conducting interlayer for the realization of optoelectronic devices. Specifically, the Authors are interested into optimizing the performances of a LED device employing the NiOx interlayer as electron blocking surface.
The manuscript has clear scientific aims and presents a very good characterization of the material in interest but following issues should be addressed
1. Minor issue: Equation 2 is just the Bragg formula that would be worth to mention
2. Minor issue: to discuss eq. (3) is useless to mention again that lambda is the x-ray wavelength
3. Major issues in table 1:
- the first and third columns must present units
- the third column fourth column should present the parameter “a” and in brackets (Lattice constant) (A)
- In this table as in table 2 and in the whole paper a discussion on uncertainty is completely missing. This is not appropriate for delivering scientific measurements. Since Authors display measurements of length like d and D with 4 decimals a discussion on the precision of 1/10000 must be presented.
4. Minor issue: all figures should present the same size of labels (a) (b) for the panels and the same should be performed for Fig. 9
5. Minor issue: I would recommend changing the notation for y-axis of Fig.5 and use a scientific one like Authors do for Fig. 8
6. Minor issue: it would be desirable that the x range in panels of Fig.9 were the same.
Author Response
Referee 1:
In the submitted manuscript the Authors discuss on the fabrication and physical properties of NiOx films made through ultrasonic spray pyrolysis technique. A series of characterizations like morphology, transport measurements and photonic absorption are conducted to assess the use of NiOx as hole conducting interlayer for the realization of optoelectronic devices. Specifically, the Authors are interested into optimizing the performances of a LED device employing the NiOx interlayer as electron blocking surface.
The manuscript has clear scientific aims and presents a very good characterization of the material in interest but following issues should be addressed
Comments and responses
- Minor issue: Equation 2 is just the Bragg formula that would be worth to mention
The name of the Bragg formula has been mentioned in Line 134.
2.. Minor issue: to discuss eq. (3) is useless to mention again that lambda is the x-ray wavelength
The repeated mention that lambda is the x-ray wavelength was removed in Line 137
- Major issues in table 1:
- the first and third columns must present units
The units of the first and third columns are now presented.
- the third column fourth column should present the parameter “a” and in brackets (Lattice constant) (A)
Now, it has been written “a (Lattice constant) ()”
- In this table as in table 2 and in the whole paper a discussion on uncertainty is completely missing. This is not appropriate for delivering scientific measurements. Since Authors display measurements of length like d and D with 4 decimals a discussion on the precision of 1/10000 must be presented.
In table 1, we have corrected the precision of the parameters to 3 decimals, because according to the XRD measurement equipment, the precision in is of 1/1000.
In table 2, we have included the uncertainty in the thickness measurements, based on the uncertainty in the refractive index, given in Line 186.
In table 3, the values of resistivity and carrier density with 3 decimal is because the Hall effect measuring system gives the measurement data results with 3 digits after point.
- Minor issue: all figures should present the same size of labels (a) (b) for the panels and the same should be performed for Fig. 9
The size of the labels has been standardized in all figures, and now the labels (a) (b) are shown in Fig. 9.
- Minor issue: I would recommend changing the notation for y-axis of Fig.5 and use a scientific one like Authors do for Fig. 8
The y-axis of Figure 5 has been put in scientific notation
- Minor issue: it would be desirable that the x range in panels of Fig.9 were the same.
Another EL spectrum has been presented in order to put the same x range (500-700 nm) for both, EL and PL, spectra in Figure 9

Reviewer 2 Report
The manuscript reports on Li-doped NiO thin films for light-emitting diodes.
The comments and questions to authors are following:
1) Equation 4 can be used if the refractive index is known. How was it determined? Note that the refractive index may vary with wavelength and Li-doping level.
2) How was the absorption coefficient obtained? The used optical model must be referred.
3) How was the average transmittance calculated?
4) Is it Tauc plot in Fig. 5? The figure caption should be corrected. The legend of Y-axis should be corrected too.
5) In Table 2, the values of resistivity and carrier density with 3 digits after point. Is it really an accuracy of Hall measurements so high?
6) What is an actual Li concentration in the films? What is a solubility limit of Li into the NiO lattice?
7) Line 245. How was the ZnS:Mn layer prepared? The thickness of ZnS:Mn must be shown.
8) Which technique was used to deposit the ZnO:Al layer?
9) Line 274: “The threshold voltage of the p-n junction…” What’s this? Is the forward I-V reproducible after prolong biasing? Was the electroluminescence observed under the reverse bias conditions?
10) Lines 275-277. Please explain in details about the tunneling current in fabricated structure at forward biasing. Note that the forward current is up to 300 mA, so it can be limited by the series resistance of the diode.
Author Response
Referee 2:
The manuscript reports on Li-doped NiO thin films for light-emitting diodes.
The comments and questions to authors are following:
Comments and responses
- Equation4 can be used if the refractive index is known. How was it determined? Note that the refractive index may vary with wavelength and Li-doping level.
Please see the answer to these questions in Lines 115-117 and Lines 186-187. The reference [23] was added to support the answer.
- How was the absorption coefficient obtained? The used optical model must be referred.
In Lines 196-197, it is specified how the absorption coefficient was obtained, using the Beer´s law.
- How was the average transmittance calculated?
Please see the answer in Line 198.
- Is it Tauc plot in Fig. 5? The figure caption should be corrected. The legend of Y-axis should be corrected too.
Yes Figure 5 shows the Tauc plots. Both, figure caption and legend of Y-axis, have been corrected according to this.
- In Table 2(3), the values of resistivity and carrier density with 3 digits after point. Is it really an accuracy of Hall measurements so high?
Yes, the Hall effect measuring system gives the measurement data results with 3 digits after point.
- What is an actual Li concentration in the films? What is a solubility limit of Li into the NiO lattice?
Please see the answer to these questions in Lines 151-156. The references [47] and [48] were added to support the answers.
- Line 245. How was the ZnS:Mn layer prepared? The thickness of ZnS:Mn must be shown.
Please see the answer to these questions in Lines 252-255. The reference [55] was added to support the answer.
- Which technique was used to deposit the ZnO:Al layer?
Please see the answer to these questions in Lines 243-245.
9) Line 274: “The threshold voltage of the p-n junction…” What’s this? Is the forward I-V reproducible after prolong biasing? Was the electroluminescence observed under the reverse bias conditions? Please see the answer to these questions in Lines 284-286.
10) Lines 275-277. Please explain in details about the tunneling current in fabricated structure at forward biasing. Note that the forward current is up to 300 mA, so it can be limited by the series resistance of the diode
A more detailed explanation about this is given in Lines 289-293.

Round 2
Reviewer 2 Report
I am not satisfied by authors’ response to my comments 1, 2 and 5.
Comment 1: Equation 4 can be used if the refractive index is known. How was it determined? Note that the refractive index may vary with wavelength and Li-doping level.
Response: Please see the answer to these questions in Lines 115-117 and Lines 186-187. The reference [23] was added to support the answer.
According to Ref. 23, the refractive index is not constant (see Fig. 2 in Ref. 23). Besides, the refractive index may vary from that for NiO single crystal likely due to voids. So, the refractive index depends on the used deposition technique. Besides, it also must vary due to observed doping-induced widening of the optical gap.
Comment 2: How was the absorption coefficient obtained? The used optical model must be referred.
Response: In Lines 196-197, it is specified how the absorption coefficient was obtained, using the Beer´s law.
It is not a suitable method for thin-films. The appropriate methods are proposed in following articles:
[1] Swanepoel, R. Determination of the thickness and optical constants of amorphous silicon. J. Phys. E Sci. Instruments 1983, 16, 1214.
[2] Emilio Márquez et al. Optical Characterization of H-Free a-Si Layers Grown by rf-Magnetron Sputtering by Inverse Synthesis Using Matlab: Tauc–Lorentz–Urbach Parameterization, Coatings 2021, 11, 1324. https://doi.org/10.3390/coatings11111324
The transmission curves of the glass substrate should be added to plots in Fig. 4 to evaluate the film non-uniformity [2]. Besides, the glass absorption in the UV is a limiting factor for determination of the optical constants at short wavelengths. Consider the use of fused silica substrates.
Comment 5: In Table 2(3), the values of resistivity and carrier density with 3 digits after point. Is it really an accuracy of Hall measurements so high?
Response: Yes, the Hall effect measuring system gives the measurement data results with 3 digits after point.
The Hall system indicates the signal with an accuracy of the voltage measurements cross the sample contacts. However, the accuracy of Hall data are limited by errors in measurements of sample thickness, magnetic field, and temperature. Which factor was dominant in your case?
Author Response
Dear editor and reviewer 2
Thank you very much for the time devoted to read our first revised version of the article, and valuable comments to improve more the quality of the paper. We have made significant changes and corrections to the manuscript based on your comments and suggestions and we hope you will be satisfied with these changes. The changes are yellow marked and with red letters in the revised version of the manuscript
Please find below our responses to the comments.
Referee 2:
I am not satisfied by authors’ response to my comments 1, 2 and 5.
Comment 1a: Equation 4 can be used if the refractive index is known. How was it determined? Note that the refractiveindex may vary with wavelength and Li-doping level.
Response a1: Please see the answer to these questions in Lines 115-117 and Lines 186-187. The reference [23] was added to support the answer.
Comment 1b
According to Ref. 23, the refractive index is not constant (see Fig. 2 in Ref. 23). Besides, the refractive index may varyfrom that for NiO single crystal likely due to voids. So, the refractive index depends on the used deposition technique.Besides, it also must vary due to observed doping-induced widening of the optical gap.
Comment 2a: How was the absorption coefficient obtained? The used optical model must be referred.
Response: In Lines 196-197, it is specified how the absorption coefficient was obtained, using the Beer´s law.
Comment 2b:
It is not a suitable method for thin-films. The appropriate methods are proposed in following articles:
[1] Swanepoel, R. Determination of the thickness and optical constants of amorphous silicon. J. Phys. E Sci.Instruments 1983, 16, 1214.
[2] Emilio Márquez et al. Optical Characterization of H-Free a-Si Layers Grown by rf-Magnetron Sputtering by InverseSynthesis Using Matlab: Tauc–Lorentz–Urbach Parameterization, Coatings 2021, 11, 1324. https://doi.org/10.3390/coatings11111324
The transmission curves of the glass substrate should be added to plots in Fig. 4 to evaluate the film non-uniformity [2].Besides, the glass absorption in the UV is a limiting factor for determination of the optical constants at short wavelengths.Consider the use of fused silica substrates.
Responses to comments 1b and 2b: Some paragraphs and references [1] and [2] suggested by the referee have been added to the article, to mention some of the most accurate methods to determine the thickness and optical constant of thin films. See Lines 194-199
In this second revised version, we have used a more appropriate method (described in reference [51] ) to determine the thickness and energy band gap of the films, including the dependence of the refractive index with wavelength. Equation (4) and (5) were removed in this second version and changed by new equation (4) on which the more accurate method is based. See the paragraphs in Lines 199 – 223, and the new Figure 5.
The transmission curve of the glass substrate was added to plots in Fig. 4, which shows that it is not a limiting factor to determine the optical constants at short wavelengths.
Comment 5a: In Table 2(3), the values of resistivity and carrier density with 3 digits after point. Is it really an accuracy of Hall measurements so high?
Response: Yes, the Hall effect measuring system gives the measurement data results with 3 digits after point.
Comment 5b:
The Hall system indicates the signal with an accuracy of the voltage measurements cross the sample contacts. However,the accuracy of Hall data are limited by errors in measurements of sample thickness, magnetic field, and temperature.Which factor was dominant in your case?
Response to comments 5b: The temperature and magnetic field is accurately controlled in the Ecopia HMS-3000 system used for Hall measurements at room temperature. The accuracy of Hall data is limited mainly by the errors in the film thickness. The Hall measurements were carried out with the values of thickness obtained though the transmittance fitting method which has specific square mean errors for each sample. Based on this, we reduced the accuracy of the values of resistivity, carrier density and mobility shown in table, to 1 digit after point.

Round 3
Reviewer 2 Report
no comments